# Anti-Inflammatory Drugs as Anticancer Agents

**DOI:** 10.3390/ijms21072605

**Published:** 2020-04-09

**Authors:** Silvia Zappavigna, Alessia Maria Cossu, Anna Grimaldi, Marco Bocchetti, Giuseppe Andrea Ferraro, Giovanni Francesco Nicoletti, Rosanna Filosa, Michele Caraglia

**Affiliations:** 1Department of Precision Medicine, University of Campania “Luigi Vanvitelli”, 80138 Naples, Italy; silvia.zappavigna@unicampania.it (S.Z.); alessiamaria.cossu@biogem.it (A.M.C.); grim.anna@tiscali.it (A.G.); marco.bocchetti92@gmail.com (M.B.); michele.caraglia@unicampania.it (M.C.); 2Biogem Scarl, Institute of Genetic Research, Laboratory of Molecular and Precision Oncology, 83031 Ariano Irpino, Italy; 3Multidisciplinary Department of Medical and Dental Specialties, University of Campania, “Luigi Vanvitelli”, Plastic Surgery Unit, 80138 Naples, Italy; giuseppe.ferraro@unicampania.it (G.A.F.); Giovannif.nicoletti@unicampania.it (G.F.N.); 4Department of Science and Technology, University of Sannio, 82100 Benevento, Italy; 5Consorzio Sannio Tech-AMP Biotec, 82030 Apollosa, Italy

**Keywords:** cancer, COX-2 inhibitors, embelin, inflammation-associated cancer, 5-LOX inhibitors, NSAIDs

## Abstract

Inflammation is strictly associated with cancer and plays a key role in tumor development and progression. Several epidemiological studies have demonstrated that inflammation can predispose to tumors, therefore targeting inflammation and the molecules involved in the inflammatory process could represent a good strategy for cancer prevention and therapy. In the past, several clinical studies have demonstrated that many anti-inflammatory agents, including non-steroidal anti-inflammatory drugs (NSAIDs), are able to interfere with the tumor microenvironment by reducing cell migration and increasing apoptosis and chemo-sensitivity. This review focuses on the link between inflammation and cancer by describing the anti-inflammatory agents used in cancer therapy, and their mechanisms of action, emphasizing the use of novel anti-inflammatory agents with significant anticancer activity.

## 1. Introduction

Inflammation is strongly related to cancer and plays a key role in tumor development and progression. It is now clear that chronic inflammation promotes carcinogenesis by inducing proliferation, angiogenesis and metastasis and reducing the response to the immune system and chemotherapeutic agents [1]. A microenvironment rich in inflammatory cells, growth factors and DNA-damage-promoting agents contributes to sustained and enhanced cell proliferation and survival, therefore promoting neoplastic risk [2]. Several epidemiological studies have demonstrated a strong correlation between inflammation and cancer. At their genesis, tumor cells are phenotypically similar to inflammatory cells since they express cytokines, chemokines and their receptors. The persistent secretion of these inflammatory mediators can induce tissue and DNA injury that leads to an accumulation of mutations in epithelial cells by promoting their growth. Mutated cells continue to produce cytokines and recruit inflammatory cells by generating a tumor inflammatory microenvironment, that contributes to angiogenesis, migration and metastasis. Inflammatory mediators were found to be more expressed in tumors than in normal tissues [2]. Thus, the use of anti-inflammatory agents, either alone or in combination with the chemotherapeutic agents, is essential for the prevention and treatment of cancer. Many anti-inflammatory agents, including NSAIDs (non-steroidal anti-inflammatory drugs), are able to interfere with tumor microenvironment by reducing cell migration and increasing apoptosis and chemo-sensitivity. In subjects undergoing to long-term NSAID therapy, a low incidence of primary or recurrent tumors was recorded. Moreover, mortality was significantly reduced in cancer patients after combination therapy with NSAIDs [3].

This review discusses the link between inflammation and cancer by describing the role of the main inflammatory mediators in tumorigenesis, angiogenesis and metastasis. In addition, the anti-inflammatory agents used in cancer therapy, and their mechanism of action will be described. The review will focus on the future perspectives regarding the use of novel anti-inflammatory agents and the related mechanisms at the basis of their significant anticancer activity. 

## 2. Inflammation

Inflammation is a physiologic process activated afterwards microbial pathogen infection, and/or wound healing. In response to tissue damage, neutrophils are rapidly recruited in the inflammatory sites by the activated endothelium and macrophages and mast cells present in the tissues through the secretion of specific mediators. Neutrophils represent the first effectors of the inflammatory response and are recruited by a four-step mechanism including L-, P-, and E-selectin-mediated activation to promote cell rolling along the vascular endothelium, leukocyte integrin activation, neutrophil immobilization on the vascular endothelium and transmigration to the inflammatory sites [1,2,3,4].

Once activated, macrophages are responsible for the production of growth factors and cytokines that attract several types of inflammatory cells to the inflamed sites. All these effectors of the inflammatory response are involved in sustaining the defense against injury.

The inflammation response is self-limiting and its duration is regulated by several molecules with a dual activity that is both pro-inflammatory and anti-inflammatory [4]. One of these molecules is the anti-inflammatory mediator TGF-β (transforming growth factor-β) that is secreted in response to the phagocytosis of apoptotic cells and contributes to the resolution of inflammation through a rapid clearance of inflammatory cells [4]. If the inflammatory response lasts too long it might shift to chronic inflammation, characterized by the presence of lymphocytes and macrophages with abnormal morphology that continuously secrete growth factors and cytokines. The persistent production of inflammatory mediators can lead to tissue and DNA damage by generating a microenvironment that promotes cell proliferation and predisposes to cancer [2].

## 3. Inflammation and Cancer

Cancer can be related to several etiologic factors including environmental stress and genomic instability [5]. Cancer development is a multi-step process, firstly initiated by genetic alterations induced by viral or chemical carcinogens and successively promoted by the exposure to chemical irritants, hormones or inflammatory mediators that induce cell proliferation and reduce the DNA repair process [5]. Finally, cells acquire a growth advantage and transform to malignant cancer cells with unregulated proliferation and enhanced angiogenesis [5].

Several epidemiological and clinical trials demonstrated a positive correlation between inflammation and cancer [4]. For instance, ulcerative colitis and Crohn’s disease [6] can increase the neoplastic risk and this process is, on the other hand, reduced by the use of anti-inflammatory agents for colitis [7]. Moreover, inflammation is often induced by microbial agents or chemical irritants; in fact, *Helicobacter pylori* infection or hepatitis B and C viruses can predispose to cancer [8].

Chronic inflammation is strictly related to cancer risk since it is characterized by an increased cell proliferation and reduced DNA repair [9]. In this context, macrophages and other leukocytes present in inflammatory sites secrete a great amount of reactive oxygen species (ROS) and mutagenic agents against microbial agents that induce a persistent tissue damage and cause DNA alterations [10]. Moreover, macrophages and T lymphocytes can produce tumor necrosis factor-α (TNF-α) and macrophage migration inhibitory factor (MIF) that interfere with the p53- and Rb-E2F pathways, contributing to tumorigenesis [11,12]. The shift from initiated cells to malignant cells requires many genetic and epigenetic events also related to chronic inflammation. Chronic inflammation is characterized by a continuous tissue and DNA injury that leads to an accumulation of mutations in epithelial cells (Figure 1) [13].

Mutated cells are able to generate a tumor inflammatory microenvironment [2] rich in macrophages, neutrophils, eosinophils, dendritic cells, mast cells, and lymphocytes that play a key role in inflammation-associated cancers [1,2]. In particular, tumor-associated macrophages (TAM) can promote tumor progression through the secretion of specific factors such as cytokines (IL-10) and growth factors (vascular endothelial growth factor (VEGF), endothelin-2, and urokinase-type plasminogen activator) that contribute to the angiogenesis [13] and suppress the immune response. Moreover, TAMs produce metalloproteinases (MMP-2 and MMP-9) that degrade the membrane basement by promoting cell invasion and metastasis [13]. Also, mast cells and tumor-associated neutrophils potentiate tumor progression by releasing cytokines and growth factors that are involved in angiogenesis, invasion and metastasis [14]. These cytokines secreted in tumor sites are specific signals to recruit lymphocytes but their specific role in tumor development is under investigation [15,16,17].

## 4. The Key Mediators of Inflammation

There are two pathways that link inflammation and cancer: extrinsic and intrinsic (Figure 2). The first is activated by inflammatory stimuli that increase the risk of cancer, the second by genetic alterations that cause cancer and inflammation. These pathways are interconnected by the secretion of inflammatory cytokines that activate specific transcription factors such as NFkB. Once activated, NFkB leads to the secretion of inflammatory mediators, growth factors, metalloproteases that contribute to the development of inflammatory tumor microenvironment [18]. Several cell components of the inflammatory process play a key role in cancer development and progression. For example cytokines, growth factors or differentiation factors that are involved in the regulation of the proliferation and differentiation of immune cells contribute to cancer by activating cell proliferation and inhibiting apoptosis of damaged cells [19,20] through several molecular signaling cascades.

### 4.1. Cytokines and Chemokines

Cytokines represent a large group of proteins secreted by the cells in response to an altered homeostatic environment that interact with specific receptors on other cells, influencing their functions. Specifically, cytokines are classified into two classes: anti-inflammatory cytokines, such as IL-4, IL-10, IL-13, IFN-α and TGF-β, and pro-inflammatory cytokines, such as IL-1β, IL-6, IL-15, IL-17, IL-23 and TNF-α [21].

The tumor necrosis factor α (TNF-α) is a cytokine involved in systemic inflammation and is a member of a group of cytokines that stimulate the acute phase reaction. It is mainly produced by macrophages, but also by CD4+ T lymphocytes, NK cells, neutrophils, mast cells, eosinophils and neurons. The effect of TNF-α appears to be very significant in the early stages of carcinogenesis (angiogenesis and invasion), inducing disease progression. Although TNF-α is considered to be a prototype of pro-inflammatory cytokines, the evidence suggests a dual role in carcinogenesis. This cytokine is recognized by two receptors: TNF-αR-1, ubiquitously expressed, and TNF-αR-2, mainly expressed in immune cells. There are controversies regarding the role of TNF-α in cancer. High concentrations of this cytokine can induce an antitumor response in a mouse model of sarcoma. Serious toxic side effects, such as hypotension and organ failure, have been associated with the systemic administration of TNF-α. Local administration has been shown to be safer and more effective, in clinical trials evaluating gene therapy with TNF-α-expressing adenoviruses, combined with chemotherapy [22]. Conversely, low levels of TNF-α production can induce a tumor phenotype. A tumor promotion mechanism by TNF-α is based on the activation of the NFkB pathway and the generation of reactive oxygen species (ROS) and reactive nitrogen species (RNS), which can induce DNA damage, thereby facilitating tumorigenesis. A study analyzed carcinogenesis associated with TNF-α using a normal human ovarian epithelial organoid exposed to prolonged doses of TNF-α. This model demonstrated the generation of a precancerous phenotype with structural and functional changes, such as tissue disorganization, loss of epithelial polarity, cell invasion and overexpression of tumor markers. According to these results, the pro- or antitumor response of TNF-α within the tumor microenvironment depends not only on the local concentration but also on its expression site in the tumor. Patients with elevated TNF-α levels in islets of non-small cell lung cancer, limited primarily to macrophages and mast cells, show higher survival rates, while patients with increased TNF-α stromal content show lower survival rates [23]. 

Another pro-inflammatory cytokine with a typical pro-tumorigenic effect is IL-6. Serum IL-6 levels in patients with systemic cancers have been shown to be elevated compared to controls in healthy patients or in patients with benign diseases. IL-6 has been proposed as a predictor of malignancy, with a sensitivity and specificity of approximately 60%–70% and 58%–90% respectively [24]. IL-6 plays a key role in promoting the proliferation and inhibition of apoptosis, by binding to its receptor (IL-6Rα) and co-receptor gp130 (glycoprotein 130), thus activating the Janus kinases (JAK) and signal transducers and transcription activators (STATs) signaling pathway, including STAT1 and STAT3. The latter belongs to a family of transcription factors closely associated with tumorigenic processes. Some studies have highlighted the effect of the IL-6/JAK/STAT signaling pathway on cancer initiation and progression [25]. IL-6 can induce tumorigenesis through the hypermethylation of tumor suppressor genes. Recently, preclinical and clinical studies showed that IL-6 was strictly related to colon cancer; it stimulated colony formation of cancer cells in vitro and its suppression inhibited tumor growth in vivo [26]. IL-6 has a role in the development of multiple myeloma, as demonstrated by its ability to induce apoptosis by blocking the IL-6R/STAT3 pathway in vitro and the resistance of the IL-6 -/- mouse to the induction of plasmacytoma. Like TNF-α, IL-6 facilitates tumor development by promoting the conversion of non-cancerous cells into cancer stem cells. In particular, the secretion of IL-6 by non-cancerous stem cells, under low attachment culture conditions, up-regulates the gene expression of Oct4 by activating the IL-6R/JAK/STAT3 signaling pathway [22].

TGF-β is a potent pleiotropic cytokine with immuno-suppressive and anti-inflammatory properties. In physiological conditions, TGF-β has a proven role in embryogenesis, cell proliferation, differentiation, apoptosis, adhesion and invasion. Three forms have been identified: TGF-β1, TGF-β2 and TGF-β3. TGF-β binds to the related type II receptor (TGF-β RII), inducing phosphorylation of the TGF-β type I receptor (TGF-β RI) and thus leading to the formation of a heterotetrameric complex that activates SMAD transcription. The SMAD transcription factors are structurally formed by a linker region, rich in serines and threonines, that connects two homologous MAD regions. Differential phosphorylation of these amino acid residues contributes to various cellular functions, including: cytostatic effects, cell growth, invasion, synthesis of the extracellular matrix, arrest of the cell cycle and migration. Therefore, the differential phosphorylation of SMAD2 and SMAD3 from TGF-β receptor activation promotes their translocation in the nucleus, where they form a complex with SMAD4, then binding to DNA where, associated with other transcription factors, they favor or inhibit gene activation [22]. The role of TGF-β in cancer is complex and paradoxical, varying from cell type and stage of tumorigenesis. In the early stages, TGF-β acts as a tumor suppressor, inhibiting the progression of the cell cycle and promoting apoptosis. Later, TGF-β increases invasion and metastasis by inducing epithelial–mesenchymal transition (EMT). There is an important evidence showing that TGF-β signaling changes are involved in human cancers. An increase in mRNA and TGF-β protein has been observed in gastric cancer, non-small cell lung cancer and colorectal and prostate cancers, and TGF-β receptor deletion or mutations have been associated with colorectal, prostate, brain and bladder cancers, in correlation with a more invasive and advanced carcinoma, with a higher degree of invasion and poor prognosis. In the tumor microenvironment, common sources of TGF-β are represented by tumor and stromal cells, including immune cells and fibroblasts. The bone matrix is also an abundant source of TGF-β and a common site for metastasis in many tumors, in correlation with tumor promotion and the invasive effects of these cytokines [22].

Interleukin 10 is known to be a powerful anti-inflammatory cytokine. Almost all immune cells produce IL-10 including T cells, B cells, monocytes, macrophages, mast cells, granulocytes, dendritic cells and keratinocytes. Cancer cells can also secrete IL-10, as can tumor-infiltrating macrophages. When IL-10 binds to its receptor it activates on the cytoplasmic side the tyrosine kinases Jak1 and Tyk2, which phosphorylate an intracellular domain of IL-10R, allowing the interaction of this with STAT1, STAT3 and STAT5, and favoring the translocation of STATs in the nucleus and the induction of target gene expression. Several studies have found that IL-10 has both pro- and anti-tumor effects. The IL-10 inhibits NF-kB; therefore, this may downregulate the expression of pro-inflammatory cytokines and act as an anti-tumor cytokine. In addition, IL-10 can exert anti-tumor activity in gliomas, melanomas, and brain and ovarian tumors, through a mechanism that involves the down-regulation of MHC-1, thus inducing cell lysis of tumor mediated by NK. Thanks to its immunosuppressive effect on dendritic cells and macrophages, IL-10 can attenuate antigen presentation, cell maturation and differentiation, allowing cancer cells to circumvent the mechanisms of immunosurveillance. Furthermore, as previously described for the IL-6, STAT3 can also be activated by IL-10, although the contradictory responses of the cytokines are determined by the receptor and by the time of activation of STAT. In particular, IL-6 leads to a transient, rapid decline in phosphorylation and nuclear localization of STAT3, while IL-10 induces sustained phosphorylation of STAT3. Through the activation of STAT3, IL-10 can also have a pro-tumorigenic effect, mediated by an autocrine-paracrine loop that involves the up-regulation of Bcl-2 and the activation of resistance to apoptosis. Similarly, elevated IL-10 levels are associated with poor prognosis [27].

Other important inflammatory mediators are chemokines, chemiotactic small cytokines that express their action by binding specific receptors expressed on endothelial cells or immune system cells. Chemokine production is stimulated by cytokines; they are able to control leukocyte infiltration in the tumor, regulate immune angiogenesis and act as growth factors [28]. Chemotactic factors are involved in cancer promotion. Several studies showed that chemokines stimulate cell growth and metastasis and induce angiogenesis in various tumors. In particular, chemokines can induce tumor cell migration by increasing the expression of MMPs that degrade extracellular matrix in a similar manner they stimulate leukocyte migration in inflammatory sites [29].

### 4.2. NFkB Transcription Factor

It has been shown that various inflammatory and carcinogenic agents, including TNF-α, cigarette smoking, lipopolysaccharides (LPS), interleukins (IL-1) and hydrogen peroxide (H2O2), activate NFκB, a nuclear transcription factor involved in tumorigenesis, inflammation, proliferation, carcinogenesis and apoptosis [30]. NF-κB comprises a family of conserved and structurally related proteins including RelA/P65, Rel/cRel, RelB, NF-κB1/p50 and NF-B2/p52. When inactivated, NFκB is found in the cytosol bound to an IκB inhibitor protein (IκBα). Through the involvement of membrane receptors, a variety of extracellular signals can activate the IκB kinase (IKK) enzyme. IKK, in turn, phosphorylates IκBα protein leading to its ubiquitination and degradation by the proteasome [31]. In this way, NFκB is activated and moved to the nucleus where it binds to specific DNA sequences called response elements (RE). This mechanism leads to a change in cell functions, such as the production of pro-inflammatory cytokines. NFκB, also, activates transcription of the coding mRNA for its IκB inhibitor subunit, thus generating a negative feed-back circuit. NFkB is activated by several cytokines as well and plays a key role in inflammatory process. In cancer, it is often constitutively activated and able to induce survival and promote cancer progression through the activation of genes coding for proteins that regulate the progression of the cell cycle (e.g., Ciclina D, c-myc) and apoptosis (e.g., CIAP, A1/BFL1, Bcl2, c-Flip) [30,31]. It is responsible for the secretion of ROS that cause DNA damage and prevent mutated cells from being destroyed. This mediator strictly links inflammation and cancer since it produces cytokines, growth factors and adhesion molecules that have pro-cancer effects. In fact, several in vitro and in vivo studies showed the involvement of NFkB in cancer promotion. For instance, knockout of IKK leaded to NFkB inactivation and decreased tumor growth in mouse model of colitis-associated cancer [32].

### 4.3. iNOS and NO Secretion

iNOS is an enzyme that catalyzes NO production and is overexpressed in several chronic inflammatory processes and cancers. iNOS is activated by pro-inflammatory cytokines or NFkB and induces DNA damage, reduces DNA repair and promotes cancer development [33]. Moreover, it stimulates angiogenesis and metastasis and can induce COX-2, an important mediator in the link between inflammation and cancer [34]. It has been reported that iNOS inhibitors were able to reduce tumorigenesis in vivo [33].

### 4.4. LOX and COX Pathways

Leukotrienes (LT) have been recognized among the various mediators of a wide range of inflammatory and allergic reactions such as rheumatoid arthritis, intestinal inflammatory diseases, psoriasis, allergic rhinitis although their primary pathophysiological implication is related to bronchial asthma [35]. Their biosynthesis requires a cell activation that stimulates the conversion of arachidonic acid into biologically active messengers. In the presence of an external stimulus, phospholipase A2 releases arachidonic acid from the membrane phospholipids. This intermediate can be attacked by two different enzymes: cyclooxygenases (COX-1 and COX-2) leading to the formation of prostaglandins (PG) and thromboxane A2 (TXA2) or lipoxygenases (5-, 8-, 12-, 15-LOX) responsible for the synthesis of leukotrienes [36]. Human 5-LOX is predominantly present in mature leukocytes including granulocytes, monocytes/macrophages, mast cells, lymphocytes B and dendritic cells in which the ability to express the enzyme is acquired during cell maturation [36]. Numerous evidences suggest the involvement of the 5-LOX pathway in the proliferation and survival of tumor cells [37,38,39,40,41,42,43,44]: (I) the enzymes necessary for the biosynthesis of LTs, as well as the LTs receptors, are present or even over-expressed in transformed cells or neoplastic tissues; (II) a substantial formation of 5-LOX products occurs at these sites; (III) the addition of 5-LOX products from the outside stimulates the proliferation and survival of tumor cells; (IV) pharmacological or genetic 5-LOX inhibition inhibits tumor cell growth and induces apoptosis. 5-LOX was abundantly detected in human or animal cancer cell lines such as brain [37], breast [38], colon [39], renal [40], mesothelium [41] esophageal mucosa [42], pancreas [43], and prostate [44], and in most of these studies there is also a concomitant increase in 5-LOX products. Recent studies [45] have shown that the expression of 5-LOX in papillary thyroid carcinoma (PTC) promotes carcinogenesis by the induction of metalloproteinases (MMPs). These enzymes, activated by both 5-LOX and its product, 5-hydroxyoxyacetanic acid, are able to degrade and remodel the extracellular matrix, promoting cell invasion. While 5-LOX uses arachidonic acid for the formation of leukotrienes, prostaglandin H synthase (PGHS) provides conversion to prostaglandin (PG) and thromboxane (TX) A2 [46]. PGHS is an enzyme that, like 5-LOX, catalyzes two coupled reactions: an initial cyclo-oxygenation and a subsequent hydroperoxide formation. However, since drugs that inhibit prostaglandin formation generally inhibit the first cyclooxygenation reaction, prostaglandin H synthase is functionally and pharmacologically described as cyclooxygenase (COX) [34]. Among these enzymes there are two isoforms:COX-1, constitutively expressed in many cells and mainly involved in the prostanoid physiological production;COX-2, whose expression is often induced in cells during inflammatory stages and therefore considered involved in pathological processes.

However, recent studies have shown that such enzymes are not only expressed during inflammatory processes, but also have a constitutive role that would explain the side effects of selective COX-2 inhibitors on the cardiovascular apparatus [34,35].

In particular, COX-2 can be activated by several inflammatory stimuli and is involved in cancer development. It is responsible for prostanoid synthesis; the effects of prostanoids are physiologically mediated by GPCRs receptors that play a role in cancer progression. Several receptors were implicated in colon cancer carcinogenesis and acted as activators of ERK or WNT pathways; the use of prostanoid receptor antagonists showed encouraging results on tumor growth inhibition [34,35].

### 4.5. Jak/Stat Pathway 

STAT3 (Signal transducer and activator of transcription 3) protein is a constitutive transcription factor expressed in several human tumors such as multiple myeloma, leukemia, lymphoma, breast cancer, prostate cancer, squamous cell carcinoma of head and neck [47]. Once activated, STAT3 goes into phosphorylation and subsequent homodimerization. After phosphorylation by some janus-like kinases (JAKs) or some soluble tyrosine kinases belonging to the Src kinase family, STAT3 homodimerizes. The dimer moves into the nucleus, binds to DNA and stimulates the transcription of some genes involved in the oncogenesis process, such as apoptosis inhibitors (Bcl-2, Bcl-xl and survivin), cell cycle regulators (cyclin D1) and inducers of angiogenesis (VEGF). STAT3, thus, plays an extremely important role in the various processes involved in development and progression of the tumor [47]. Moreover, it can be stimulated by several cytokines.

### 4.6. MAPK Pathway 

Mitogen-activated protein kinases (MAPKs) regulate cell growth, differentiation, survival and immune and stress responses [48,49,50]. Their activation occurs through consecutive phosphorylations, in which each passage is regulated by different kinases. These phosphorylative events can be inactivated by specific phosphatases. There are three pathways mediated by MAPKs: ERK1/2, c-JUN N-terminal kinase 1, 2 and 3, (JNK1/2/3) and p38 MAPK α, β, Paths δ and γ. These pathways are triggered by cytokines, growth factors, hormones and osmotic stress [51,52,53,54,55,56,57,58]. Once activated, JNK as well as ERK1/2 and p38 moves to the nucleus to regulate the expression of the genes involved in growth and proliferation, such as c-Myc, c-Jun, Jun-B, EIK, P53 and NFAT [59].

### 4.7. Phosphoinositide-3-Kinase (PI3K) Pathway

Posphoinositide-3-kinase (PI3K) phosphorylates phosphatidylinositol-4,5-bisphosphate, after having received the upstream signals, in phosphatidylinositol-3,4,5-triphosphate (PIP3) which acts in turn on the PREX1/2, Akt and protein kinase 1 (PDK1). AKT, once activated by PDK1, acts on numerous pathways involved in survival, cell growth, protein synthesis and metabolism. The reaction can be negatively regulated by some phosphatases, PTEN, SHIP1/2 and INPP4B [60]. Several studies have shown the association between some mutations in genes coding for P110α, P110β, the regulatory subunit P85α and, PTEN, SHIP1/2 and INPP4B (PI3K inhibitory phosphatase) and malignancy occurrence [61]. When PTEN is mutated, PI3K is activated and induces high levels of programmed death ligand 1 (PD-L1) while PI3K inhibitors increase immune cell response to tumors [62]. Patients with pancreatic cancer showed mutated K-Ras, responsible for a constitutive activation of PI3K/Akt signaling pathway [63].

### 4.8. CREB Signaling Pathway

CREB plays a very important role in cell survival, in the differentiation of neurons and in metabolism. In particular, CREB-mediated gene expression is essential for cell survival induced by the nerve growth factor (NGF), which promotes the survival of sympathomimetic neurons, through the activation of downstream target genes [64]. The increase in cell survival can also be induced by Bcl-2, responsible for NGF activation, independently of CREB [65]. The transcriptional activity of CREB is induced by reversible phosphorylation to serine residues, from various kinases, such as protein kinase A (PKA), protein kinase B (PKB/AKT), mitogenic kinase (MAPK) and 90 ribosomal kD S6 Kinase [66]. It plays a key role in the development of resistance against the inhibitors of the Raf-MEK-ERK and PI3K/AKT signal pathways [67,68]. An increased expression and activation of CREB is associated with tumor progression, resistance to chemotherapy and reduced survival of patients [65]. This is mediated by aberrant activation of components of the relevant cAMP signaling pathways, such as G-coupled, tyrosine kinase receptors and cytokines/JAK/STAT pathway, but also downstream signaling pathways. In particular, an increased expression of CREB was found in patients with acute lymphoid and myeloid leukemia [69], in clear cell soft tissue sarcomas (CSST) [70], non-small lung carcinoma (NSCLC) [71], glioblastoma [72], mammary carcinoma [73], melanoma [74] and diffuse malignant mesothelioma when compared to adjacent normal tissues [75].

### 4.9. Wnt/Beta Catenin Pathway

The Wnt family consists of secreted glycolipoproteins that regulate cell proliferation, cell polarity and cell fate determination during embryonic development and tissue homeostasis [76]. Wnt signaling is regulated by the transcriptional co-activator β-catenin that controls key developmental gene expression programs [77]. Extracellular Wnt proteins activate the canonical signaling pathway dependent on β-catenin through the commitment of rippled co-receptors (FZD) and the protein related to the low-density lipoprotein receptor (LRP) or the non-canonical pathway independent of β-catenin through various receptors such as FZD, RTKs and tyrosine kinase-like orphan receptors (RORs). In the absence of specific Wnt ligands, the β-catenin present in the cytosol is degraded by the destruction complex consisting of three different proteins, the antigen-presenting cell (APC), casein kinase I (CKI), axin (AXIN), glycogen synthase kinase 3 (GSK3B). Specifically, this complex binds to the cytosolic β-catenin and allows the sequential phosphorylation of the latter from CKI (to S45) and GSK-3 (to S33/S37/T41). The phosphorylated β-catenin is then recognized and ubiquitinated for the degradation by the proteasome. Two serine/threonine phosphatases, protein phosphatase 1 (PP1) and protein phosphatase 2A (PP2A), regulate the Wnt signaling pathway by binding Axin complex, APC, GSK3 and CK1 to inhibit their interaction with the β-catenin [78]. However, alterations of the genes that code for various components of Wnt/β catenin signaling, such as Wnt, FZD, APC and LRP5/6, have been reported in numerous studies. The aberrant constitutive activation of the signaling of Wnt/β-catenin caused by mutations in genes, such as APC or CTNNB1 (coding β-catenin), is involved in the tumorigenesis of many organs, including the intestine, stomach, liver, ovaries and pancreas [79]. Alterations in the genes encoding APC have been described in the development of specific colorectal tumors [80]. The involvement of the canonical Wnt pathway in head and neck squamous cell carcinoma (HNSCC) has also been reported in several studies [81]. The Wnt/β-catenin signaling pathway interacts with many other signaling pathways, including NF-κB, Smad3, Notch, forkhead box O (FOXO) and hypoxia-inducing factor-1α (HIF-1α), extending the spectrum of biological functions of this pathway [82].

## 5. Role of Inflammatory Mediators in Tumorigenesis

As described below, we have analyzed the main factors and cytokines involved in tumorigenesis by focusing on ROS production, epithelial–mesenchymal transition (EMT), angiogenesis and metastasis.

### 5.1. ROS and RNS Production Associated with Inflammation 

In an inflammatory response, the activation of epithelial and immune cells triggers the generation of ROS and RNS, respectively through the induction of NADPH oxidase and nitric oxide synthase (NOS). NADPH oxidase is one complex protein composed of several membrane-associated subunits that catalyze the superoxide anion (O^2−^), leading to the production of peroxide of hydrogen (H_2_O_2_) mediated by superoxide dismutase (SOD-). On the other hand, NOS generates nitric oxide (NO) from L-arginine, which can be converted into RNS such as nitrogen dioxide (NO_2_), Peroxynitrite (ONOO-), and nitrogen trioxide (N_2_O_3_). Different of NOS are produced according to the cell type: inducible NOS (iNOS) in phagocytes and constitutively in endothelial and neuronal cells (eNOS and nNOS). ROS and RNS have a powerful antimicrobial role in phagocytes and also act as second messengers in signal transduction. Activation of phagocytes can directly induce reactive oxygen and nitrogen species (collectively called RONS), activating NOX2, NADPH oxidase, and iNOS. Furthermore, TNF-α, IL-6, and TGF-β trigger the generation of RONS in non-phagocytic cells. The increased expression of NADPH oxidase and NOS and their RONS products has been identified in several cancers, suggesting that free radicals have a role in the genesis and malignant progression. Elevated RON levels have been observed in various chronic inflammatory diseases, such as *H. pylori*-associated gastritis and inflammatory bowel disease (IBD), suggesting a role in cancer risk. Several mechanisms have been proposed to clarify the participation of RONS in cancer development. RONS induce cellular oxidative stress and damage to lipids, proteins, and DNA, as well as the production of 8-oxo-7, 8-dihydro-2′-deoxyguanosine (8-oxodG), currently used as a marker of damage to DNA. Identifying these DNA damage markers in chronic inflammatory processes, such as gastritis associated with *H. pylori*, hepatitis, and ulcerative colitis, underlines the relevance of RONS in pathologies with an increased risk of cancer. An increase in iNOS, 3-nitrotyrosine, and 8-oxodG has been found in the livers of patients with primary sclerosing cholangitis. In addition, RNS interferes with DNA repair, as is demonstrated in iNOS-overexpressing cells that are unable to repair modified 8-oxodG. RONS are generated by cellular stress and by modification of macromolecules, although they are also involved in the regulation of signaling pathways, such as cell survival and proliferation through Akt, Erk1/2, and the activation of hypoxia-inducible factor 1 (HIF-1) [22].

### 5.2. Tumor Growth Associated with Inflammation

As repeatedly stressed, inflammation is important in generating malignancy through the exposure of pro-inflammatory cytokines and the sustained activation of signaling pathways such as NF-kB and STAT3. After the transformation in the malignant cells, these cytokines are also involved in tumor growth by stimulating the proliferation of neoplastic cells and eluding immunosurveillance. Several cytokines have growth factor activity. In one study it was noted that the silencing of TNF-α in a cell line of the gallbladder cancer decreased cell proliferation and invasion by an autocrine effect, influencing the signaling pathways of TNF-α/NF-kB/AKT/ Bcl-2 in these cells [22]. The pro-tumorigenic role of IL-17 has also been implicated in other types of cancer. Mice with carcinogenic-induced uterine tumors deficient in the IL-17 receptor showed a lower tumor incidence and reduced tumor size. Other molecules have been reported in cancer that can influence tumor growth through regulation of the IL-6/STAT3 signaling pathway. Inflammatory mediators such as Hmgbl, IL-23 and IL-17 can promote tumor growth by activating the IL-6/STAT3 pathway in a mouse model of melanoma. In cholangiocarcinoma, a high expression of the oncoprotein, gankirin, promotes tumor proliferation, invasion and metastasis through the activation of the IL-6/STAT3 signaling pathway [83].

### 5.3. Epithelial–Mesenchymal Transition (EMT) Associated with Inflammation

The epithelial–mesenchymal transition (EMT) is a process in which, following a chronic stimulus, epithelial cells with basal–apical polarity lose their phenotype and acquire the characteristics of non-polarized and migrating mesenchymal cells [84]. Molecular markers have been identified to assess whether or not an epithelial cell has gone through the EMT process. The main marker is the loss of E-cadherin, an event associated with the destruction of cell–cell junctions. E-cadherin (epithelial cadherin) is necessary for the formation of strong and stable adherent junctions and therefore for the maintenance of the epithelial phenotype and normal tissue architecture of the adult. A reduction in the expression of E-cadherin, as has been shown in various malignancies, plays a crucial role in the loss of cell differentiation and in dissemination. It can be said that E-cadherin appears to be the custodian of the epithelial phenotype. N-cadherin (neural cadherin) is normally found only in cells of the nervous system, but is produced in some carcinoma cells that have lost the expression of E-cadherin and, in this cellular context, is associated with an increased invasive potential [85]. A relevant inflammatory mediator in EMT is TGF-β and extensive evidences support the concept that EMT can be induced by pro-inflammatory cytokines. TNF-α and IL-6 may urge the TGF-β signaling pathway through the progression of EMT. Both cytokines promote the activation of NF-kB, which regulates the expression of the transcription factors involved in EMT, coordinating the effects of Snail1, Snail2, Twist, ZEB1 and ZEB2. Furthermore, IL-6 induces cell invasiveness in EMT, through the increased expression of vimentin (it is the main component of the cytoskeleton of mesenchymal cells) and the down-regulated expression of E-cadherin, both mediated by signaling JAK/STAT3/Snail, as shown in head and neck cancer. Finally, ROS production can promote EMT: therefore, exposing kidney epithelial cells to ROS induces the expression of TGF-β, the SMAD signaling pathway, and EMT [22].

### 5.4. Angiogenesis Associated with Inflammation

Angiogenesis includes the processes that lead to the generation of new blood vessels from an already existing vascular network. This angiogenic process is important in tumor development because new blood vessels penetrate and supply nutrients and oxygen to cancer cells. Several angiogenic factors secreted by tumor cells have been identified, in particular the vascular endothelial growth factor (VEGF), which is expressed in response to cytokines and growth factors. Furthermore, the characterization of tumor associated macrophages (TAM) obtained from metastatic lymph nodes (MLN) in an animal model of melanoma, has shown that MLN are mainly constituted by infiltrating macrophages TIE2/CD31. This subpopulation overexpresses in a significant way VEGF and is directly related to angiogenesis. Some studies have shown that TNF-α may have a double-edged role in angiogenesis, depending on the used doses. High doses of TNF-α inhibit angiogenesis in mice while low doses promote vascularization of the area. The anti-angiogenic effect of TNF-α is linked to the down-regulation of ανβ3 (adhesion molecule) and of the angiotensin signaling pathway, while pro-angiogenic responses have been associated with the increased expression of VEGF, VEGFR, IL-8, and FGF. Therefore, low levels of TNF-α increase tumor growth, induce angiogenesis of several tumors in mice, and stimulate a sub-population of tumor-associated myeloid cells and the coexpression of endothelial and myeloid markers with pro-angiogenic/pro-vascular properties. The tumor source of TNF-α can be derived from myeloid or tumor cells and through autocrine activation can stimulate tumor growth and angiogenesis. Another important angiogenic factor is IL-6, which induces the expression of VEGF in a dose-dependent manner in gastric cancer cell lines. Similarly, IL-6 promotes angiogenesis through activation of the STAT3 pathway in cervical cancer. Together, the secretion of IL-6 and the subsequent phosphorylation of STAT3 are involved in the up-regulation of angiogenic mediators, such as VEGF, HIF1α, the VEGFR2 co-receptor and neuropilin 2 (NRP2) [22].

### 5.5. Metastases Associated with Inflammation 

Metastases are malignant cells that detach themselves from the primary tumor and spread to other organs where they can reproduce and generate new tumors. The metastatic cascade is modulated by the action of several cytokines released by surrounding cells such as tumor associated macrophages (TAMs), tumor infiltrating lymphocytes (TIL), and cancer associated fibroblasts (CAFs), promoting the escape of cancer cells and dissemination; the influence of TNF-α has been studied in various experimental animal models. The administration of this cytokine leads to a significant increase in the number of lung metastases. In some studies, it has been proposed that cancer cells activate myeloid cells to generate a favorable microenvironment for metastasis. In Lewis lung cancer (LLC) cells, high levels of IL-6 and TNF-α have been induced in bone marrow-derived macrophages. TNF-α -/- and non-IL-6 -/- mice injected with LLC cells showed increased survival and a reduction in lung tumor multiplicity, suggesting a key role of TNF-α in LLC metastasis. IL-6, in turn, is up-regulated in several cancers and is implicated in the ability of cancer cells to create bone metastases. In contrast, IL-10 exhibits anti-tumor function. The replacement of IL-10 to the human cell line of melanoma A375P, which does not produce endogenous IL-10, using a vector containing the murine IL-10 cDNA, reversed tumor growth and lung metastasis. This result states that IL-10 production by cancer cells inhibits metastasis. There is also a strong relationship between EMT and metastasis, suggesting that, in the early stages of the metastatic cascade, EMT allows for the migration and invasion of cancer cells. For this reason, the inflammatory mediators involved in EMT, in particular TGF-β, could play an important role in promoting metastasis [22].

## 6. Inflammation as Target for Cancer Prevention

Since inflammation can predispose to tumor, targeting inflammation and the molecules (COX-2 cyclooxygenase 2, NF-kB, VEGF,) involved in inflammatory process could represent a good strategy for cancer prevention and therapy [1,2,3,4], (Table 1).

### 6.1. NSAIDs 

Cancer prevention by NSAIDs mostly works by acting on the pathway of the eicosanoids. In the past, several clinical studies have demonstrated that NSAIDs had anti-tumor activity [114,115] and their long-term use reduced the incidence of colorectal, esophageal, breast, and lung cancers [116]. NSAIDs showed toxicity and non-specific effects that were lower than those induced by conventional chemotherapy and were able to inhibit tumor progression by interfering with tumor inflammatory microenvironment [89].

Several studies demonstrated that anti-inflammatory agents could increase apoptosis and sensitivity to the conventional therapies and decrease invasion and metastasis [89,117,118,119], making them useful candidates for cancer therapy.

Epidemiological data showed that the incidence and mortality for colorectal and lung cancer patients that used NSAIDs was lower than for those that did not use these drugs [120,121]. These results are well documented for colorectal cancer but not really clear for other cancers. Several studies investigated the association between aspirin or NSAIDs and breast cancer risk and only half of these studies confirmed the potential use of NSAIDs to reduce breast cancer risk. On the other hand, in other studies the long term use of aspirin did not decrease the risk of developing breast cancer [122] and no promising results were recorded in terms of cancer prevention.

FAP (familial adenomatous polyposis) patients showed a decreased recurrence and lower polyp number after the use of Sulindac [123,124]. Other NSAIDs, such as ibuprofen and piroxicam, were able to reduce breast and colorectal cancer risk by showing a significant correlation between anti-inflammatory agent use and decreased cancer incidence [90,125]. Epidemiological data regarding the correlation of NSAID use and the risk of pancreas, prostate, bladder and renal cancers are controversial and limited. Some studies reported an increased risk of hematologic malignancy and kidney cancer in NSAID users. The antitumor effects of NSAIDs depend on doses, duration, cancer type. In most of epidemiological studies, study populations had different baseline characteristics that complicate the interpretation of the results and limit the findings [126,127].

In particular, a meta-analysis that included 300,000 participants with prostate, breast, lung, and colorectal cancer from 16 studies suggested the potential of NSAIDs to reduce distant metastasis in different types of cancer. NSAIDs could represent optimal candidates for cancer therapy, unfortunately, the significant side effects of NSAIDs limit their use. Among the NSAIDs, anti-inflammatory agents with more specific activity against COX-2 were developed in order to decrease adverse effects. Clinical studies demonstrated that FAP patients that used a specific COX-2 inhibitor, celecoxib, displayed regression of existing adenomas [124]. On this basis, celecoxib was approved by the FDA for the use in the adjuvant therapy of FAP [101]. Although these are promising results, the use of celecoxib remains controversial, because many studies did not support its antitumor effects depending on its adverse effects including GI toxicity and cardiotoxicity. Other COX2 inhibitors such as rofecoxib (VioxxR) and valdecoxib (BextraR) are under investigation [128]. Moreover, clinical phase I and II trials are ongoing but at the moment, these drugs are used as adjuvant agents not as monotherapy because of their toxic effects.

### 6.2. Corticosteroids

Corticosteroids are useful to counteract the adverse effects of chemo-and radio-therapy but several studies showed that they have also significant anti-cancer effects both alone or in combination. For instance, lung cancer incidence was reduced in mice exposed to tobacco smoke after treatment with dexamethasone [129]. Moreover, the anticancer activity of conventional drugs was potentiated by dexamethasone administration in several animal models of cancers [106,130]. Preclinical studies demonstrated a significant efficacy of dexamethasone in inhibiting renal cancer cell growth and breast cancer progression. Other glucocorticoids, such as prednisone and hydrocortisone, showed significant antitumor effects in vitro and in vivo [131,132].

## 7. Anti-Cancer Effects of Anti-Inflammatory Agents

Several preclinical and clinical studies demonstrated that the combination between chemotherapeutic drugs and anti-inflammatory agents was effective in improving patient prognosis. Monotherapy is not able to completely eradicate cancer but anti-inflammatory drugs are useful adjuvants for conventional therapeutic strategies.

The mode of action at the basis of the anti-inflammatory drug antitumor effects is not completely defined but three different potential mechanisms have been described.

### 7.1. Chemoprotection 

One of the problems related to the conventional anticancer therapies is the side effect profile. Chemotherapy often induces toxicity to both tumor and several normal tissues, reducing patient quality of life. Several studies showed that the combination between conventional therapies and anti-inflammatory agents could decrease the side effects of chemotherapeutics. For example, the concomitant administration of celecoxib and docetaxel to patients with metastatic prostate cancer reduced toxicity to bone marrow [133] and the combination of celecoxib and FOLFIRI (folinic acid, fluorouracil and irinotecan) or capecitabine decreased diarrhea episodes [99,134]. Recently, the GECO (Gemcitabine-Coxib) study evaluated the effects of refecoxib in combination with gemcitabine in patients with NSCLC and showed that refecoxib use for 3 months ameliorated the quality of life [121].

Recent studies focused on dexamethasone, commonly used as anti-emetic, and demonstrated that it was able to reduce hematologic toxicity induced by gemcitabine and carboplatin. Another glucocorticoid, budesonide, in combination with irinotecan decreased diarrhea episodes [135,136].

Other anti-inflammatory agents such as aspirin can act as anti-thrombotic drugs, decrease arterial thrombosis by facilitating the passage of chemotherapeutics and ameliorate prognosis.

Anti-inflammatory drugs such as COX-2 inhibitors or dexamethasone reduced neurotoxicity by inhibiting the expression and activity of matrix metalloproteinases (MMPs) 3 and 9 and VEGF, therefore stabilizing blood-brain barrier [137].

### 7.2. Alterations in Pharmacokinetics or Metabolism

Anti-inflammatory agents are able to modify the pharmacokinetics of the other drugs. In fact, several studies showed that dexamethasone decreased hematologic toxicity of conventional therapies (gemcitabine, carboplatin and doxorubicin) in mice, probably by changing drug pharmacokinetics [106,130]. Glucocorticoids induced no significant differences in plasma pharmacokinetics but altered the uptake of gemcitabine or carboplatin by the spleen and bone marrow and increased the amount of drugs that reached the tumor [130]. The combination between dexamethasone and adriamycin showed similar results [106]; glucocorticoids can interfere with the pharmacokinetics of conventional drugs by increasing their anticancer effects and decreasing their side effects.

In addition, anti-inflammatory agents can alter the metabolism of chemotherapeutic drugs; for instance, rofecoxib acts as CYP1A2 inhibitor [138] and induces changes in concentration, half-life and clearance of the other drugs that are metabolized by CYP1A2. Dexamethasone induces CYP2D6 activity and celecoxib inhibits it [139], thus interfering with the efficacy of tamoxifen, a substrate of CYP2D6. Diclofenac inhibits the glucuronidation of DMXAA by blocking its metabolism and increasing its plasma concentration [140]. The tumor interstitial fluid pressure (IFP) is responsible for the decreased uptake of chemotherapeutic drugs into the tumor site and minor chemoresponsivity [141]. Several studies demonstrated that VEGF and PDGF antagonists, colecoxib and dexamethasone reduced IFP, thus increasing the amount of drugs that reached the tumor [100,142]. Anti-inflammatory agents can alter metabolism of other drugs, improving their efficacy and decreasing their toxicity, but it is important to consider that cytochrome p450 enzymes that play a key role in metabolism of the major part of the drugs can be polymorphic and lead to unexpected results in terms of toxicity and efficacy of the drugs.

### 7.3. Chemosensitization

In addition to the ability of anti-inflammatory drugs to decrease toxicity of conventional drugs by altering their metabolism, the effective anticancer effects of the combination between anti-inflammatory agents and chemotherapeutic drugs are probably due also to the chemosensitization by anti-inflammatory agents. Several preclinical studies showed that the combinations between celecoxib and etoposide, doxorubicin, vincristine or irinotecan were additive or synergistic in vitro. In particular, celecoxib in combination with irinotecan or doxorubicin induced tumor growth inhibition in rats models of neuroblastoma [103]. Moreover, celecoxib increased the response of prostate cancer cells to docetaxel in vitro and in vivo and sensitized gliomas to radiation [143].

Dexamethasone was also able to increase the efficacy of carboplatin and gemcitabine in several xenograft models probably interfering with pharmacokinetics [106,130]. In addition, dexamethasone is an immunosuppressive drug that blocks cytokines secretion and reduces lymphocyte proliferation by protecting bone marrow and spleen from the action of chemotherapeutic drugs that target proliferating cells.

Several studies have demonstrated that NSAIDs induced apoptosis in different tumor types by directly acting on NFkB pathway. Aspirin and sulindac are able to sequester NFkB in the nucleolus and block the transcription of NFkB targets, such as cytokines, growth factors, adhesion molecules, etc. [86]. Celecoxib increased the anticancer activity of doxorubicin probably increasing IkB expression thus inhibiting NFkB activity [144]. Also, dexamethasone was able to suppress NFkB pathway [145].

Anti-inflammatory agents are able to induce apoptosis in cancer cells also interfering with proteins involved in programmed cell death. For example, celecoxib potentiated the efficacy of docetaxel by activating caspases and PARP and decreasing XIAP activity [103]. Aspirin acted on caspases and pro-apoptotic proteins [87] and probably inhibited NFkB activity by blocking proteasome [88]. Dittmann et al. demonstrated that celecoxib sensitized cancer cells to radiation by inhibiting EGFR in an independent manner by COX-2 [146]. Many anti-inflammatory agents retain their antitumor activity also if they do not act as COX-2 inhibitor. Another mode of action of NSAIDs is the ability to inhibit drug resistance molecules; sulindac or COX-inhibitors inhibited the P-glycoprotein (P-gp) expression and activity [147,148]. On the other hand, celecoxib increased expression of multidrug resistance proteins MRP4 and MRP5 and decreased the anticancer activity of conventional drugs [148]. New clinical trials that evaluate the use of several anti-inflammatory agents as monotherapy or in combination in cancer therapy are under investigation.

## 8. Novel Anti-Inflammatory Drugs with Anti-Cancer Activity

Several clinical trials evaluating the use of anti-inflammatory agents in combination with chemotherapeutic drugs for cancer prevention and therapy have been performed and anti-inflammatory agents showed encouraging results in terms of efficacy and toxicity. Preclinical and clinical studies that evaluate the anti-cancer effects of new anti-inflammatory agents and their mode of action are ongoing.

### 8.1. Anti-Cancer Agents Based on COX-2 Inhibitors

Several studies that evaluated the use of COX-2 inhibitors in cancer therapy showed that the antitumor effects of these agents were independent from their ability to inhibit COX-2, therefore new agents that retained the antitumor activity but had minor side effects compared with COX-2 inhibitors were developed [149].

Several novel agents based on celecoxib did not act as specific COX-2 inhibitors but showed a significant efficacy and decreased toxicity in preclinical studies. In particular, these new agents induced in vitro and in vivo cancer growth inhibition through different mechanisms compared to COX-2 specific agents. They were able to induce anoikis, cell cycle arrest by targeting AKT, MAPK or STAT3 pathways [149,150].

### 8.2. NO-Donating NSAIDs

NO donating NSAIDs represent analogs of NSAIDs with decreased side effects. These agents retain the active component able to induce anti-inflammatory effects linked by a spacer to NO. The potency of NO-NSAIDs composed of aromatic spacers was higher compared to those with aliphatic ones. NO, once released, protects GI from injury induced by the active drug by reducing GI toxicity [151]. Diclofenac, naproxen, aspirin, sulindac, ibuprofen have been modified in order to obtain NO donating molecules [152]. Several NO-donating NSAIDs showed significant anticancer effects in preclinical studies; they induced apoptosis, growth inhibition or cell cycle arrest on different types of cancer in vitro and in vivo [151,152]. All the preclinical studies about NO-donating NSAIDs showed a higher efficiency compared to NSAIDs but additional studies are required to better understand their potential in cancer prevention [151]. NO-ASA (NO-acetylsalicylic acid) showed synergic or additive effects in combination with oxaliplatin or 5-fluoruracil on colon cancer models and reduced cancer risk in model of induced pancreatic cancer. Moreover, the ortho and para-isomers of NO-ASA were more effective than meta-isomers in inhibiting cell growth and survival in colon cancer. The use of NCX4016 (NO-acetylsalicylic acid) for colorectal cancer prevention was evaluated in a phase I study but the possible genotoxicity of this agent induced the premature termination of the study [152].

### 8.3. Natural Products 

Different foods or natural products have shown anti-inflammatory effects such as grapes/red wine (resveratrol), garlic (various compounds), curry powder (curcumin) [153,154,155,156,157]. Moreover, these natural compounds showed anti-cancer effects on cancer cells or xenografts due to the induction of apoptosis or cell cycle arrest.

These compounds act as anti-inflammatory agents by targeting NFKB, MAPK and JNK pathways or inhibiting VEGF or COX enzymes and probably these effects contribute to their anticancer activity [153,154,155]. Recently, clinical studies showed that the use of natural products as adjuvant agents for conventional therapies gave encouraging results for patients. The combination between curcumin and vinorelbine or 5-fluorouracil was synergistic in inhibiting cancer cell proliferation. Ginseng saponins increased the response of cancer cells to chemotherapeutic drugs and reduced hematologic toxicity after radiotherapy [155]. Garlic compounds were able to potentiate the anticancer effects of cytarabine and fludarabine in myeloid leukemia cells [156] and increased the response of prostate cancer cells and xenograft tumors to docetaxel with decreased side effects [157]. The combination of natural products with the conventional anti-inflammatory agents increased their efficacy and decreased their toxicity [158]. In fact, garlic compound S-allylmercaptocysteine potentiated the in vitro effects of sulindac on cell growth inhibition and apoptosis induction [158]. Further investigations showed that Barberine, present in plants of the genera Coptis, acts as anti-inflammatory, anticarcinogenic, and proapoptotic agent via inhibition of transcription factor NFκB and downregulation of COX-2 [42]. It inhibited COX-2 transcriptional activity in colon cancer cells at concentrations higher than 0.3 µM [39]. Moreover, Barberine was able to reduce the metastatic potential of melanoma cells by inducing COX-2 inhibition and reactive oxygen species (ROS) production which in turn increased AMPK phosphorylation [45].

### 8.4. LOX Inhibitors 

To confirm the implication of 5-LOX in the pathophysiology of cancer, many researchers have applied the use of pharmacological instruments such as 5-LOX inhibitors (Zileuton, ZYflo, ABT-761) [159,160,161], FLAP inhibitors (MK-886) [162,163] or LTA4 hydrolase [164] and LTs antagonists (Zafirlukast and Montelukast) [159,165,166] in blocking cell proliferation and inducing apoptosis in vitro and in vivo. Despite strong potential for anti-LTs therapy in cancer prevention and treatment, so far few clinical trials have been conducted to evaluate the efficacy of 5-LOX inhibitory drugs in antitumor therapy. It has been shown that treatment with LOX inhibitors in some pancreatic cell lines significantly reduced cell proliferation [159]; however, this contrasted with the simultaneous administration of 5-HETE and 12-HETE metabolites [159,160,161]. The potential use of LOX inhibitors in the prevention and treatment of pancreatic cancer has also been demonstrated by in vivo studies with athymic mice in which LOX inhibitors led to a decreased tumor growth [160].

A study by Gosh showed that 5-LOX inhibition by MKAP1, a FLAP inhibitor, induced apoptosis in LNCaP and PC3 cell lines [163].

5-LOX inhibitors resulted inducers of apoptosis in esopharyngeal cancer cells [164] and 12-LOX inhibitors had antiproliferative and pro-apoptotic effects on gastric cancer cells [167]. Finally, few studies on hematopoietic cells showed that the 5-LOX AA861 inhibitor induced apoptosis in the P388 leukemia cell line [168]. Natural inhibitors, derived from plants, have been tested to evaluate their effectiveness in blocking the synthesis of LTs in cells isolated from animal or human organs. In 1981, a study was published on the NDGA, a polyphenolic derivative extracted from a mexican plant (Larrea divaricata) as the first 5-LOX inhibitor of natural origin [169]. Later on, other compounds were isolated from the Chinese plant Artemisia Rubris: caffeic acid, eupatiline and 4-dimethylnaphthylin inhibited the activity of 5-LOX both directly on purified enzyme and in mastocytoma cell assays [170]. In the same year, esculetine, present in many plants, was identified as a 5-LOX inhibitor [171]. A recent study by Weiz showed that the embelin acted as a 5-LOX inhibitor because it could interfere with the arachidonic acid (AA) metabolism by blocking the activity of 5-LOX [172].

### 8.5. Embelin and Its Derivatives

Embelin is a benzoquinone derivative able to interfere with the arachidonic acid metabolism, whose structure consists of a polar 2,5-dihydroxy-1,4-benzoquinone nucleus linked at position 3 to a long hydrophobic alkyl chain that gives solubility in the apolar phase and allows the molecule to cross the cellular barrier. Thanks to a study carried out in collaboration with prof. Oliver Werz (University of Jena, Germany), it has been highlighted that this molecule can block the activity of both 5-LOX and mPGES-1 with IC50 values of 0.06 and 0.2 μM respectively [172]. The chinonic structure of the membrane is crucial for the inhibitory activity. In fact, it has been shown that the molecule is able to inhibit the enzyme without suffering a hydroquinone reduction process. Docking studies demonstrated that embelin did not act as a chelant of iron, but it fits with its undecylated chain in the hydrophobic channel in which normally the iron-catalyzed oxygenation reaction of the AA is carried out. The benzoquinone ring co-ordinates with three amino acids (Gln363, Gln557 and Tyr181) by determining hydrogen bonds that stabilize the enzyme-molecule interaction. The binding between the embelin and 5-LOX is mediated by two water molecules that co-crystallize with the complex and form hydrogen bonds with two amino acids such as Asn425 and Thr364. Recent studies showed that embelin and its derivatives had antioxidant, anti-inflammatory, antitumor and analgesic properties [173]. In particular, they were able to induce apoptosis in human myeloid cells HL by interacting with microtubule proteins and activate caspases in pancreatitis. It has been shown that naturally occurring embelin derivatives (5-O-methyl embelin and 5-O-ethyl embelin) [174] induced cell cycle arrest of HL-60 cells at G0/G1 stage in a dose- and time-dependent manner. The potent antitumor activity of embelin is linked to various mechanisms, such as: (i) Inhibition of XIAP, (ii) Inhibition of c-FLIP expression, (iii) Activation of PPARγ, (iv) Inhibition of NF-κB, (v) Inhibition of STAT3 (Figure 3).

Several studies have demonstrated that Embelin binds to XIAP protein and induces apoptosis. A preclinical study performed on PC-3 and LNCaP prostate cancer cells that expressed high levels of XIAP, healthy human fibroblasts (WI-38) and normal prostate human epithelial cells (PrEC) used as controls confirmed the selectivity profile for cancer cells; in fact, IC50 values for PC3 and DU were lower than those of healthy cell lines. Further biological studies have demonstrated that embelin can induce apoptosis in prostate PC-3 and leukemia HL-60 cells by activating caspase 9 through down-regulation of XIAP protein [174,175].

In two recent studies, it has been shown that embelin is able to induce TRAIL-induced apoptosis both in glioblastoma cells and pancreatic cancer cells if associated with FLIP antisense oligonucleotides by down-regulating anti-apoptotic protein FLIP [176]. Various studies have shown that PPARγ activation inhibits cell growth and induces differentiation and apoptosis in colon cancer cells [177]; embelin was able to reduce cell proliferation and induce apoptosis both in HCT116 and HT-29 colon cancer cells, by increasing PPARγ receptor expression. Moreover, it showed a chemopreventive effect on mice after the induction of colorectal cancer through PPARγ upregulation. In a work carried out by Ahn and collaborators [178], the ability of the embelin to modulate NFκB pathway was evaluated. In particular, it has been observed that it sequentially inhibits the activation of the inhibitory subunit of NFκBα (IκBα kinase), IκBα phosphorylation, IκBα degradation, p65 phosphorylation, nuclear translocation and the expression of antiapoptotic genes regulated by NFkB.

In a recent work [179], the role of embelin in STAT3 pathway regulation has been demonstrated. It blocks the activation of the constitutive form of STAT3 in several tumor cell lines, such as U266 (multiple human myeloma), DU-145 (prostate cancer), SCC4 (squamous cell head-neck carcinoma). In particular, embelin selectively inhibits 5-LOX and m-PGES-1. Such enzymes are over-expressed in some inflammatory diseases such as osteoarthritis and rheumatoid arthritis [180] and in some types of cancer [46] where they contribute to tumor growth, angiogenesis, and resistance to apoptosis.

Docking studies performed on the 5-LOX crystallographic model have clarified the molecule binding mode with the target, highlighting which amino acid residues are involved in binding to the inhibitor.

Considering the chemical structure of the lead, a series of analogues of the embelin has been designed and synthesized wherein the undecyl chain has been replaced by saturated and unsaturated aliphatic chains of varying lengths or rings with one or two aliphatic or aromatic terms.

Based on the biological results on the synthesis derivatives, it was possible to obtain important structure-activity correlation information that demonstrated the importance of the length and flexibility of the alkyl chain in the receptor fitting. Our group have been involved for a long time in the synthesis and the study of biological effects at the basis of anti-cancer and anti-inflammatory activity of quinone-based compounds [181,182,183]. Taking into account that lipoxygenase (LOX) metabolites have been implicated in tumor development and progression, we evaluated the anticancer activity of embelin analogues. Our previous study has demonstrated that 3-((decahydronaphthalen-6-yl)methyl)-2,5-dihydroxycyclohexa-2,5-diene-1,4-dione induced growth inhibition, apoptosis and a cell cycle arrest together with high levels of p21 and p27. Moreover, it induced a significant cleavage of caspases 8, 9, 3 and 7, disrupted the interaction cIAP2/XIAP by degrading XIAP and inhibited NF-kB pathway.

Moreover we found that the orthoquinone EA-100C(4,5-dimethoxy-3-alkyl-1,2-benzoquinone) and its reduced form EA-100C red (3-tridecyl-4,5-dimethoxybenzene-1,2-diol), synthesized with a C13 n-alkyl chain lacking hydroxyl groups, showed IC50 values of 10 and 60 nM, respectively, in cell-free assays representing the most potent 5-LO inhibitors [184].

The anticancer activity of these novel 5-LO inhibitor derivatives have been demonstrated on glioblastoma (GBM) cells. EA-100C was able to induce apoptosis, cell cycle modulation, and autophagy; on the other hand, EA-100C red induced ER stress-mediated apoptosis associated to autophagy through CHOP and Beclin1 up-regulation and activation of caspases 3, 9, JNK and NFkB pathway [185].

### 8.6. Other Therapeutic Agents Targeting Inflammation

Statins have been developed as HMG-CoA reductase inhibitors therefore, inhibiting cholesterol synthesis. They have also shown to have anti-inflammatory and anti-cancer effects when used both as monotherapy or in combination with other anti-inflammatory or chemotherapeutic agents (5-fluorouracil, doxorubicin and cisplatin) [186,187,188].

Blocking chemokine receptors can represent a good therapeutic strategy for cancer. Several chemokines and their receptors are overexpressed in cancer and strictly related to cancer progression and metastases such as CCR7, CXCR4 and CXCR7. Preclinical studies showed that agents targeting CXCR4 receptor including antibodies, siRNAs or antagonists were able to inhibit tumor growth and metastasis. In particular, they showed encouraging results in cancer of the head and neck and inhibited angiogenesis in colon cancer [189,190]. Other therapeutic agents targeting inflammatory microenvironment or host immune response include Toll-like receptor agonists or antagonists, antibodies (rituximab, zerumbone and CXCR4), or other agents directed against pro-inflammatory molecules or their receptors.

## 9. Conclusions

Inflammation is strictly related to cancer and plays a key role in tumor development and progression. Targeting inflammation, either alone or in combination with the chemotherapeutic agents, is therefore essential for the prevention and treatment of cancers. Several studies have shown that the use of anti-inflammatory drugs is associated with decreased cancer incidence and recurrence. Many anti-inflammatory agents can be used as adjuvants for conventional therapies but additional studies are needed to better understand their potential as anticancer drugs. Early diagnosis and treatment of chronic inflammation could be useful to decrease cancer development. Anti-inflammatory agents that obtained FDA approval have a limited use in cancer therapy for the presence of off-target effects and toxicities. Therefore, combinations, changes in doses or regimens or development of new agents could allow to overcome these limitations and represent a new therapeutic strategy in cancer. It is also important to develop safe and effective agents that reduce the risk of cancer. For example, NSAIDs exhibit several side effects, so analogs with lesser adverse effects have been developed, whose anti-tumor effects are still to be determined. Cancer is a complex pathology and requires a targeted approach: knowledge of the biochemical pathways involved in cell proliferation has allowed the design of new target-based drugs. It is important to consider that these new drugs should be tested in preclinical and in vivo studies and then be transplanted into clinical use without losing their effectiveness and safety. The idea of targeting the inflammatory tumor micro-environment is innovative but a better understanding of the molecular events underlying the activity of the new synthesized agents can be helpful in developing new effective therapeutic strategies.

## Figures and Tables

**Figure 1 ijms-21-02605-f001:**
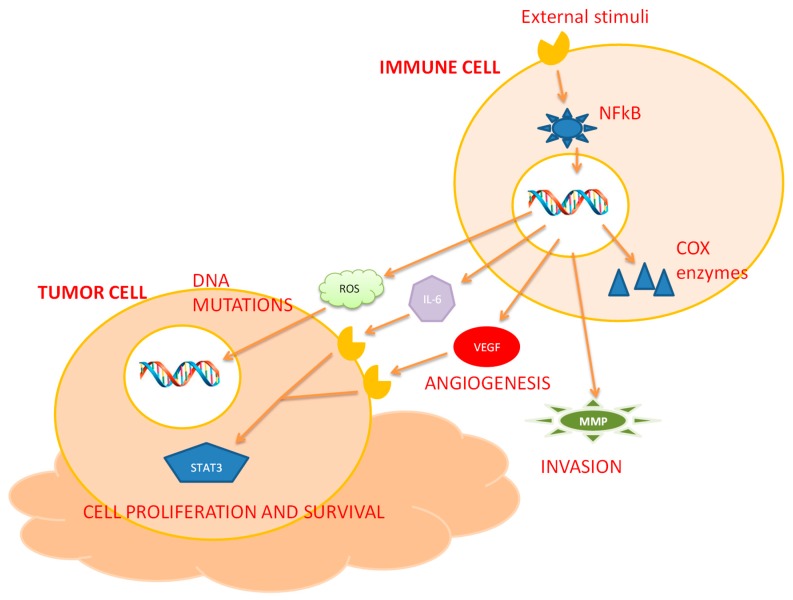
Inflammation and cancer. Various inflammatory and carcinogenic agents can activate the transcription factor NFkB. Once activated, it binds to specific DNA sequences in the nucleus and induces the production of pro-inflammatory cytokines and COX enzymes. Activated immune cells produce specific cytokines (IL-6, VEGF, etc.) and metalloproteinases (MMP-2 and MMP-9). IL-6 and growth factors can induce STAT3 activation by leading to cell proliferation and survival while metalloproteases degrade the membrane basement, promoting cell invasion. Moreover, macrophages secrete a great amount of reactive oxygen species (ROS) and mutagenic agents against microbial agents that induce a persistent tissue damage and cause DNA alterations by contributing to tumorigenesis.

**Figure 2 ijms-21-02605-f002:**
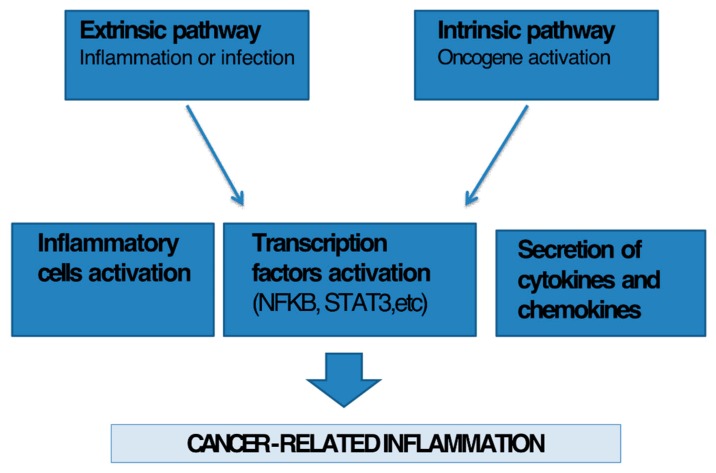
Link between inflammation and cancer. There are two pathways that link inflammation and cancer: extrinsic and intrinsic. The first is activated by inflammatory stimuli, the second by genetic alterations. These pathways are interconnected by the secretion of inflammatory cytokines that activate specific transcription factors (NFKB, STAT3, etc.) and lead to the secretion of inflammatory mediators including growth factors, metalloproteases that contribute to the development of inflammatory tumor microenvironment.

**Figure 3 ijms-21-02605-f003:**
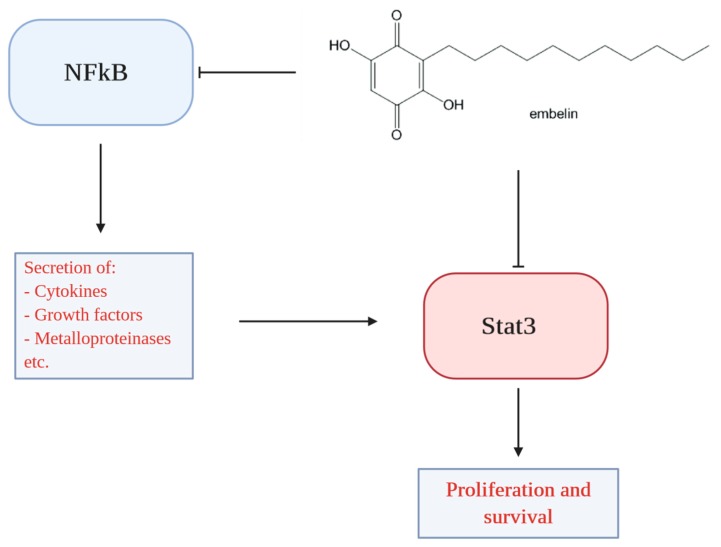
Anticancer effects of embelin. The potent antitumor activity of embelin is linked to various mechanisms, including inhibition of NF-κB and inhibition of STAT3.

**Table 1 ijms-21-02605-t001:** Preventive and anticancer effects of anti-inflammatory drugs.

Drug	Effect	Reference
Aspirin	Induced activation of NF-kB pathway in colon cancer cells	[86]
	Induced activation of caspase8/Bid and Bax pathway in gastric cancer	[87]
	Induced apoptosis in neuroblastoma cells through the inhibition of proteasome function	[88]
	Preventive effect on bladder cancer	[89]
	Preventive effect on breast cancer	[90]
	Preventive effect on colorectal cancer	[91,92]
	Preventive effect on esophageal cancer	[93]
	Preventive effect on lung cancer	[94]
Colecoxib	Induced apoptosis in prostate cancer cells	[95]
	Induced endoplasmic reticulum stress in hepatoma cells	[96]
	Inhibited the expression of survivin via the suppression of promoter activity in human colon cancer cells	[97]
	Preventive effect on bladder cancer	[98]
	Preventive effect on breast cancer	[99]
	Preventive effect on cervix cancer	[100]
	Preventive effect on colorectal cancer	[101]
	Preventive effect on lung cancer	[102]
	Preventive effect on neuroblastoma	[103]
	Preventive effect on prostate cancer	[104]
Dexamethasone	Induced cell death in multiple myeloma mediated by miR-125b expression	[105]
	Preventive effect on breast cancer	[106]
	Preventive effect on rectal cancer	[107]
	Preventive effect on multiple myeloma	[108]
Ibuprofen	Inhibited activation of nuclear β-catenin in human colon adenomas	[109]
	Preventive effect on breast cancer	[90]
Piroxicam	Prevented colon carcinogenesis by inhibition of membrane fluidity and canonical Wnt/β-catenin signaling	[110]
	Preventive effect on colorectal cancer	[111]
Sulindac	Induced activation of NF-kB pathway in colon cancer cells	[112]
	Preventive effect on breast cancer	[113]

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
