# Peer review of "Anti-Inflammatory Drugs as Anticancer Agents"

_ijms, 2020, doi:10.3390/ijms21072605_

Round 1

Reviewer 1 Report

It is a very ambitious manuscript concerning a subject of great importance in pathogenesis and cancer treatment. However in most volume the included information are too superficial. In my opinion the manuscript lack the focus. Additionally most of the information are not new (the book type). Inflammation leading to cancer it is a long process that involved a complex cell signaling pathways that are omitted here.

Author Response

We have very appreciated the comments and suggestions of the referee and we have taken advantages of his/her considerations. We have tried to answer all his/her final issues as specified below and we hope now to satisfy the criticisms.

Comments and Suggestions for Authors

Point1: It is a very ambitious manuscript concerning a subject of great importance in pathogenesis and cancer treatment. However in most volume the included information are too superficial. In my opinion the manuscript lack the focus. Additionally most of the information are not new (the book type). Inflammation leading to cancer it is a long process that involved a complex cell signaling pathways that are omitted here. 

Answer: Our manuscript is focused on novel anti inflammatory drugs that show a significant anticancer activity and their mechanism of action as we have better described in the final part of the Introduction section  (line 46). In this light, we have modified the title of the paragraph (line 643) that represents the focus of the paper in which we propose to give innovative information about the molecular mechanisms at the basis of the tumor activity of novel anti inflammatory drugs,some of them we have tested in our laboratory. According to the reviewer we have described in deep and added new information about the inflammation mechanisms that lead to cancer (lines 140, 395) and in particular cell signaling pathways that we have omitted (line 315). We hope that our efforts to improve the manuscript according to the reviewer suggestions can satisfy all the criticisms.

Thank you for your kind consideration and helpful suggestions.

Reviewer 2 Report

  1. There are several typing errors in the manuscript. 
  2. line 224 ……please delete f
  3. I suggest the authors would reorganize the table 1 and table 2. I recommend that the authors combine table 1 with table 2.
  4. The title of section 7. Future Prospective is not proper. I suggest the authors could revise the title.
  5. The authors should provide high resolution figures

Author Response

We have very appreciated the comments and suggestions of the referee and we have taken advantages of the considerations. We have tried to answer all the final issues as specified below and we hope now to satisfy the criticisms.

Comments and Suggestions for Authors

  1. Point: There are several typing errors in the manuscript. 

Answer: As suggested by the reviewer, we have corrected all the typing errors. 

  1. Point: line 224 ……please delete f  

Answer:According to the reviewer we have deleted f at the line 224. 

  1. Point: I suggest the authors would reorganize the table 1 and table 2. I recommend that the authors combine table 1 with table 2. 

Answer: As suggested by the reviewer we have reorganized tables 1 and 2 by combining them in only one table.

  1. Point: The title of section 7. Future Prospective is not proper. I suggest the authors could revise the title. 

Answer: As correctly suggested by the reviewer we have modified the title and changed “Future prospectives’ in “Novel anti-inflammatory drugs with anti-cancer activity” (line 643). 

  1. Point: The authors should provide high resolution figures 

Answer: According to the reviewer we have provided higher resolution figures 

We hope that our efforts to address the specific issues raised by the referees have indeed improved the manuscript.
Thank you for your kind and helpful suggestions.

Round 2

Reviewer 1 Report

The authors addressed and responded to all questions raised during the first evaluation of the manuscript. Thank you